# Identification of Risk Factors for Multiple Non-Melanoma Skin Cancers in Italian Kidney Transplant Recipients

**DOI:** 10.3390/medicina55060279

**Published:** 2019-06-16

**Authors:** Elisa Zavattaro, Paolo Fava, Federica Veronese, Giovanni Cavaliere, Daniela Ferrante, Vincenzo Cantaluppi, Andrea Ranghino, Luigi Biancone, Maria Teresa Fierro, Paola Savoia

**Affiliations:** 1Department of Translational Medicine, University of Eastern Piedmont, Via Solaroli, 17-28100 Novara, Italy; daniela.ferrante@med.uniupo.it (D.F.); vincenzo.cantaluppi@med.uniupo.it (V.C.); 2Department of Medical Sciences, University of Turin, 10126 Turin, Italy; fava_paolo@yahoo.it (P.F.); g.cavali_dr@yahoo.it (G.C.); andrea.ranghino@unito.it (A.R.); luigi.biancone@unito.it (L.B.); mariateresa.fierro@unito.it (M.T.F.); 3Department of Health Science, University of Eastern Piedmont, 28100 Novara, Italy; federica.veronese@med.uniupo.it (F.V.); paola.savoia@med.uniupo.it (P.S.)

**Keywords:** carcinogenesis, immunosuppression, non-melanoma skin cancer, risk factors, skin cancer, transplanted patients

## Abstract

*Background and objectives:* Non-melanoma skin cancers (NMSCs) represent the most frequently encountered malignancy in organ transplant recipients and their incidence increases proportionally to the duration of immunosuppression. Furthermore, patients of this group often develop multiple and more aggressive cancers and, to date, risk factors for the development of multiple NMSCs have not been yet established. The present study aimed to identify risk factors for multiple NMSCs in a cohort of Italian kidney transplant recipients (KTRs). *Materials and Methods:* We consecutively included all KTRs referring to two post-transplant outpatient clinics of North-Western Italy between 2001 and 2017. In this cohort, we evaluated different clinical (endogenous and exogenous) risk factors in order to establish their correlation with NMSCs. *Results:* 518 KTRs were included, of which 148 (28.6%) developed keratinocyte cancers, with a single tumor in 77 subjects, two skin cancers in 31 patients, 3 in 21 patients, whereas at least 4 NMSCs developed in 19 KTRs. We observed an increased risk of the development of cutaneous neoplasms for the male gender, old age at transplantation (>50 years), light phototype, solar lentigo, history of sunburns, or chronic actinic damage. Considering patients affected by multiple keratinocyte neoplasms, we observed a significant association of actinic damage and solar lentigo with an increased risk of NMSCs; their significance was confirmed even at the multivariable model. *Conclusions:* Our results confirm the role played by chronic cutaneous actinic damage in carcinogenesis on KTRs and highlight the significance of individualized periodic dermatological screening.

## 1. Introduction

Cancer is one of the major causes of death in solid organ transplant recipients (OTRs) and its incidence increases proportionally with the duration of immunosuppressive therapy [1]. In this group of subjects, the skin is the organ at higher risk to develop cancer and non-melanoma skin cancers (NMSCs) represent the most frequent skin cancer observed. In general terms, main risk factors for the development of NMSCs are represented by light skin, intense UV exposure, older age, beta-human papillomavirus (β-HPV) infection, and immunosuppression [1,2,3]. Hence, NMSCs are predominantly observed in fair-skinned individuals, on sun-exposed areas, and their number increases with age of patient and transplant duration. However, these factors cannot fully explain the differences in number, histotype, time of onset, and progression of NMSCs observed in OTRs.

In OTRs, squamous cell carcinomas (SCCs) are more frequent than basal cell carcinomas (BCCs) with a 4:1 ratio, while in the general population the ratio can vary from 1:2 until 1:4. Moreover, population differences in BCC and SCC rates have been reported, and this could be explained by the different populations studied (i.e., diverse geographic origins, phototype) [4].

In this study, we aimed to describe our cohort of kidney transplant recipients (KTRs) with respect to the presence of skin cancer as well as different risk factors correlated with the development of cutaneous neoplasms.

Furthermore, since NMSCs are also relatively common in the general population, we decided to focus our study on the patients developing multiple skin cancers (two *versus* three *versus* more than three), to increase the strength of the study and to properly recognize the key factors predisposing to chronic cutaneous carcinogenesis.

## 2. Materials and Methods

### 2.1. Patients

In this study, we included all kidney transplant recipients (KTRs) followed up from December 2001 to May 2017 at the post-transplant Outpatient Clinic in the Dermatology Units of the University Hospitals of Turin and Novara, two cities located in North-Western Italy.

Each patient underwent a periodic complete dermatologic examination performed by the same group of dermatologists skilled in the follow up of transplanted subjects. The frequency of clinical examination was stratified upon the patient’s risk factors for NMSC development (i.e., strong UV-exposure, previous diagnosis of skin cancer) and it generally varied between 3 and 12 months.

For each patient, we have collected all the demographic data (date of birth, gender, date and age at transplantation, age at the last visit) and gathered information regarding their life habits through a questionnaire (past and present sun and/or UV-lamp exposure, previous sunburns, use of sunscreen, outdoor job). Complete clinical data were also recorded during skin examinations (skin phototype, the presence of solar lentigo, seborrheic keratosis, cherry angiomas, cutaneous warts and/or signs of skin actinic damage, personal or family history of skin cancer, immunosuppressive regimen, previous dialytic treatment, type of kidney disease). Skin phototypes were assessed in 6 groups, according to the Fitzpatrick classification [5].

Every skin cancer developed by the patients was submitted to appropriate medical and/or surgical treatment and all the related information was recorded in the patient’s medical record, including time of onset, location (sun-exposed versus not sun-exposed areas), and histological features.

Patients that did not fulfill the questionnaire and/or did not attend all the periodic scheduled visits were not included in the present study.

The present study was performed according to the Declaration of Helsinki, and its conduction was approved by the local Ethical Committees (SS-DERMO6) (Date: 15 January 2016).

### 2.2. Statistical Analysis

The Cox proportional hazard regression models were used to analyze the factors that possibly affect skin cancer risk. The hazard ratios (HR) and 95% confidence interval (95% CI) were calculated: a value of HR > 1 indicates a higher risk in the group with a given characteristic than in the reference group.

In order to identify which of the risk factors are associated with developing multiple tumors vs. single tumors, we computed logistic regression analysis. The odds ratios and 95% confidence interval (95% CI) were calculated.

For both analyses, each variable was considered separately in a model and then we performed a multivariable model using the forward selection method. The significance of each individual variable was evaluated using the likelihood ratio test (LRT). Age at transplant and gender were included in all models as potential confounding variables.

*p*-Values less than 0.05 were considered statistically significant. Statistical analysis was performed using Stata version 14.0 (StataCorp, College Station, TX, USA).

## 3. Results

### 3.1. Patient Clinical Characteristics

A total of 518 KTRs were included in the present study, with a prevalence for males (M 332, F 186; 64.1% and 35.9%, respectively). Kidney transplantations were performed from January 1974 to June 2016, with 301 patients (58.1%) receiving a transplant before 2007, and, hence, submitted to a long period of immunosuppression (more than 10 years). The median time after transplant was 9.2 years. Mean age at transplantation was 48.3 years (M 48.9; F 47.3) and mean age at follow-up was 59.1 (M 59.4; F 58.3).

Regarding the considered risk factors, we decided to group skin phototype in three groups (I–II corresponding to light phototypes, III–IV to medium phototypes, and V–VI to dark phototypes). Data about skin phototype were available in 505/518 patients; among them, the 30.7% had a phototype I and II, and the 67.7% III and IV, in accordance with the prevailing phenotypic characteristics of the inhabitants of our geographical area.

Moreover, different kinds of kidney diseases were grouped on the basis of their origin into the following three categories: immune-mediated, congenital, and acquired diseases.

Unfortunately, given the retrospective nature of the study, data on the previous disease related to the kidney failure were available only in a subset of patients (326/518).

The results of clinical data and life habit data are depicted in Table 1 and Table 2.

### 3.2. Risk Factor for Keratinocyte Neoplasms

The Cox regression analysis was performed in order to evaluate the significant predictors of skin cancer after transplant considering the clinical characteristics and life habits (sun and/or UV-lamp exposure, previous sunburns, use of sunscreen, outdoor job) of patients with skin cancer and patients who did not develop neoplasms.

In our case series, we observed an increased risk of the development of NMSCs for the male gender, light skin, solar lentigo, history of sunburns, and evidence of chronic actinic damage. Moreover, those older than 50 years at transplantation showed an increased risk of the early development of skin carcinomas (mean 5.8 vs. 15 years; *p* < 0.0001). On the contrary, a lack of significance was observed with other considered clinical characteristics. Lack of significance was also detected for other life habits data and the development of NMSCs in our cohort (i.e., sun- and UV-lamp exposure, use of sunscreen, outdoor job).

The Cox regression analysis results are depicted in Table 3.

The multivariable model established an increased risk of skin cancer for male gender (HR 2.0; 95% CI 1.4–3.0), older age at transplantation (≥50 years vs. <50 years; HR 6.0; 95% CI 3.9–9.1), and presence of actinic damage (HR 2.2; 95% CI 1.6–3.0) (data not shown). All models were adjusted by age at transplant and gender.

In order to detect if gender was associated with the risk factors considered in the analysis, the logistic regression analysis has been conducted and only the actinic damage was at the limit of significant association (male vs. female OR = 1.6; CI 95% 1.0–2.4).

### 3.3. Type and Timing of Skin Cancers

In our cohort, patients had a mean age at the time of first tumor excision of 62.0 years (SD 10.0); mean time after transplantation was 9.0 years (SD 7.7) for the appearance of the first skin cancer, with shorter period observed for the subsequent neoplasms. In fact, the second cancer developed 2.4 years (SD 3.1) after the first (11.6 years after transplantation, SD 9.4), while the third appeared 1.6 years after the previous (SD 1.8) (13.2 after transplantation, SD 10.2) and the fourth tumor was excised after a further 1.1 year (SD 1.6) (14.2 after transplantation, SD 11.4)

In our cohort, a total number of 278 NMSCs was observed. In detail, they were categorized as follows: 164 BCCs, 114 SCCs; the SCC:BCC ratio was 1:1.4. Bowen’s disease, keratoacanthoma, and hyperkeratotic actinic keratosis were included in the SCC group, indeed they represent in situ SCC.

Furthermore, when considering the different types of tumors developed in our cohort, we observed that 79 patients had developed at least one SCC, while 102 had developed at least one BCC.

During follow-up, a total of 148 (28.6%) patients developed at least one NMSC. In our cohort, most of the patients (77) presented a single tumor, while 71 developed two or more neoplasms (in detail, 31 patients with 2 tumors, 21 subjects with 3 tumors, and 19 patients that developed at least 4 skin cancers). These data are summarized in Table 4.

In order to evaluate the risk of developing multiple tumors versus developing a single neoplasm, the logistic regression models were used. The logistic models confirmed a significant association for presence of actinic damage (OR 4.8; 95% CI 2.3–9.8) and solar lentigo (OR 3.7; 95% CI 1.7–7.8) and an association at the limit of statistical significance for male gender (OR 2.0; 95% CI 0.9–4.4). The presence of seborrheic keratosis (OR 1.3; 95% CI 0.6–2.8), cutaneous warts (OR 1.9; 95% CI 0.8–4.3), and cherry angioma (OR 1.2; 95% CI 0.6–2.4) do not reach statistical significance. These data are shown in Table 5. Actinic damage and solar lentigo remained significant in the model obtained after forward selection method (OR 4.2; 95% CI 1.9–8.9 and OR 3.0; 95% CI 1.4–6.7). All models were adjusted by age at transplant and gender.

## 4. Discussion

In the present study, we report the results based on a large cohort of KTRs referring to two dermatology post-transplant outpatient clinics in the North-West of Italy. Our work was mainly focused on the identification of risk factors for the development of multiple non-melanoma skin cancers. It is well known in the literature that the risk of cutaneous tumors is generally increased in transplant recipients [1,2,3,6,7,8,9,10]. However, in clinical practice, only a few patients develop multiple cutaneous carcinomas in the post-transplant period, while for others the risk is not significantly increased compared to the general population. In the present study, an increased risk for NMSCs was observed in male patients with fair skin, older age at transplantation, and with cutaneous signs of chronic sun-damage (including the presence of solar lentigo and previous sunburns). In subsequent analysis, the presence of chronic actinic damage, solar lentigo, and male gender confirmed their strict role in favoring the development of multiple skin cancer. Chronic sun-damage has confirmed its role also in multivariable analysis and it has emerged as a fundamental risk factor for multiple NMSCs among our patients.

In the literature, numerous reports dealing with the incidence of skin cancer in OTRs are available, often showing controversial results, depending on different populations, and/or experimental approach. In general terms, NMSCs are increased in OTRs and the average time to develop such neoplasms is estimated in a range of 4–9 years after transplantation, and our data are in line with those intervals [1,2,6,8]. Furthermore, older age at transplantation and male gender have been universally recognized as common risk factors for skin carcinomas, unless other factors could also severely promote tumor progression (i.e., some immunosuppressive drugs, different organ transplanted, long transplant duration) [6,7,9,10].

Another important factor leading to the increase of skin carcinomas development is represented by sun exposure: the cumulative UV radiation exposure seems, in fact, to be of importance in this process [11,12,13,14,15]. Indeed, it is well known that strong UV exposure is able to provoke genetic alterations in keratinocytes, thus, leading to the development of skin cancer. The biological UV effects on keratinocytes are responsible for the so-called field cancerization: a cutaneous area prone to develop NMSC, characterized by either visible (i.e., actinic keratosis, cutaneous dyschromia, and atrophy) or microscopically detected damages [16]. In addition, many studies confirm that patients with pale skin are at higher risk for NMSCs, mainly developed in sun-exposed areas of the body, thus confirming the role of UV [17,18,19]. In this regard, Gogia et al., have also reported a direct association between the risk of SCC and light skin type in solid organ transplant recipients [20].

In the present study, light phototypes were demonstrated to be generically associated with attitude to NMSCs development and almost all the subjects with cutaneous carcinomas had phototype I and II. Even if intense sun exposure (i.e., outdoor job) was not linked with increased risk of skin cancer, a history of previous sunburns (occurring more frequently in light phototypes and following strong sun exposure) was associated with NMSCs in our cohort, thus confirming the role of UV in skin carcinogenesis. On the other hand, the presence of chronic sun-damage and solar lentigo, two clinical features that represent the effect of previous strong sun-exposure, was strictly associated with more than 3-folds increased risk of developing multiple NMSCs.

Male gender, older age at transplantation, strong UV-exposure (mainly represented by occupational sun exposure and lack of sunscreen usage) and presence of actinic keratosis (AKs) were found to be related to NMSCs development in our previous study [20]. Herein, we can confirm the strong relationship between age, gender, and AKs with skin cancer in KTRs. On the contrary, in the present study cherry angiomas did not reach statistical significance, otherwise than previously observed in our series where their presence had a protective effect [21]. We are not able to properly explain such differences between the two studies, since we could expect to confirm the data, also because the populations in the two studies are from the same geographic area. However, we have to underline that cherry angiomas are benign and quite common cutaneous lesions, mainly developed in the elderly population, as it is our cohort. In addition, in our case series, we did not detect a correlation between the presence of seborrheic keratosis and cutaneous warts and the development of skin carcinomas (either single or multiple neoplasms). Lally et al., have previously reported a strong association between seborrheic warts and NMSCs as well as between seborrheic keratosis and viral warts in a cohort of KTRs, but no specific HPV type was detected and any firm conclusion was drawn [22]. In this regard, it has to be taken into account that NMSCs in immunosuppressed patients are mainly linked to a reactivation of β-HPV types, while cutaneous warts are caused by other genotypes (i.e., α, γ, μ), unless we can speculate that the high frequency of such viral lesions could be a further indicator of immunosuppression, thus favoring the skin cancer appearance. Our group has previously published a paper concerning the role of β-HPVs in early stages (namely tumor initiation and progression) of cutaneous carcinogenesis in KTRs; the presence of β-HPVs protein expression was detected in numerous types of NMSCs and also in the context of benign lesions (i.e., seborrheic keratosis) [23].

The current study examined exclusively KTRs, confirming a high prevalence of skin cancer respect to the general population. This observation finds a great deal of support in the literature, whereas data regarding other neoplasms (i.e., melanomas) show variability among different studies [24,25,26,27,28].

Based on the literature, an inverse ratio SCC:BCC is often reported in OTRs [1,18,29]; in the present study we observed a ratio 1:1.4, with a majority of BCC, that is similar to what is registered in the general population. A possible explanation for diverse SCC/BCC ratios reported in clinical studies can be linked to differences in latitude, sun-exposure habits, and phototype. In fact, other studies conducted in OTRs living in Mediterranean countries have reported a ratio ranging from 1:1.1 to 1:2.2 and comparable to that observed in our series [19,30,31,32,33].

Moreover, the percentage of patients included in the present study who developed at least a single skin cancer is in line with other reports based on KTRs [33,34].

Regarding multiple NMSCs, in our cohort, we observed a lack of association between the histological type of first and subsequent tumor. Limited data exist on the development of subsequent NMSCs in OTRs, but it has been recently shown to increase the risk of developing a further SCC in OTRs with cutaneous SCC, and this was less frequently reported for BCCs [35,36,37,38,39].

Tessari et al., did not observe a significant association between risk factors and the development of subsequent NMSCs in a cohort of Italian heart and kidney transplant recipients affected by BCC and/or SCC. Notwithstanding, they reported a high presence of AKs in their group of patients [37]. Accordingly, our study seems to confirm once again the role of chronic sun-damage (including actinic keratoses) in promoting skin cancer development.

In this regard, systemic retinoids are frequently prescribed in skin cancer prevention in high-risk NMSC patients; moreover, the regular use of sunscreen is strictly recommended in transplanted patients and/or in subjects developing multiple NMSCs [40,41,42]. Unfortunately, oral retinoids administration needs a careful follow up in terms of possible adverse events and interactions with other therapies/comorbidities affecting the patient. Moreover, in recent years, the use of mTOR inhibitors (a class of molecules that inhibit the mammalian target of rapamycin, also called rapalogs) in immunosuppression has demonstrated its effectiveness in significantly decreasing the risk of developing further NMSCs (mainly SCC) in OTRs [43,44]. Their use is currently recommended in transplanted patients that had already developed skin cancers. However, such treatments are not used as a first-line therapy, and drug-related adverse events are quite common. With this in mind, it is clear that the early recognition of risk factors is of great value and could also allow the creation, in the near future, of a specific algorithm and scores to quantify the risk of developing keratinocyte tumors. This could help in identifying which patients require the administration of oral retinoids and rapalogs.

Moreover, we have to point out that the present study is a clinical study based on direct and periodic complete skin examination, differently from those performed retrospectively through cancer registries. We believe that this represents a further strength of our study.

On the contrary, since the present study suffers from some limitations, our data deserves a thorough discussion. First, in our cohort, we grouped SCC, AK, Bowen disease, and keratoacanthoma in the same category of NMSC (excluding BCC). This is because they represent the different kinds of the same spectrum of cutaneous squamous cancers: in fact, the existence of a continuum between actinic keratosis towards SCC is universally recognized. Furthermore, the presence of actinic keratosis in this group can be argued while it is a feature of chronic sun damage. In this regard, it should be specified that we only considered the histologically-confirmed actinic keratosis as cutaneous neoplasms, namely hyperkeratotic and highly infiltrated grade III lesions, that required surgical excision. On the other side, since our study is based on both clinical and histological features, this is of great help to avoid misclassification bias and to minimize recall bias. Another limitation of the study could be represented by the exclusion of patients who did not attend all the clinical examinations and/or not provide complete clinical information. Since our study aimed to identify risk factors for the development of multiple NMSCs, it is clear that the lack of clinical data prevented any results being reached. Another point to be discussed is the fact that we did not evaluate the role of different immunosuppressive regimens and the previous dialytic treatment duration in favoring NMSC. In this regard, it is mandatory to take into account that immunosuppressant therapy is often composed of a mix of at least two or three agents and that their dosage is subject to numerous changes during the treatment course. Therefore, we believe that it could be very hard to establish an association between drugs and NMSC development. In our study cohort, since the administration of drugs with anti-proliferative action (i.e., oral retinoids, mTOR inhibitors) was limited to a small number of patients, we are not able to draw any conclusion. This could be due to the fact that our patients have a median long time of transplantation and most of them received grafts before rapalogs were used as immunosuppressive treatments. Furthermore, even immunosuppression due to the dialysis treatment, whose duration is not always easy to measure, should be considered. In our series, family history of skin cancer did not seem to play a role in cutaneous carcinogenesis, even though only a few patients reported the presence of such factor.

## 5. Conclusions

In conclusion, our study highlights the striking effect of chronic sun-damage and of the consequent cancerization field in increasing both the overall risk of NMSC and the risk of developing multiple keratinocyte tumors in KTRs. Based on these results, the role of periodic skin examination in transplanted subjects and every effort in promoting prevention against NMSCs are confirmed to be of primary importance.

## Figures and Tables

**Table 1 medicina-55-00279-t001:** Clinical characteristics of our cohort of patients.

	N° Patients (%)	With Skin Cancer (%)	Without Skin Cancer (%)
Gender			
- Male	332 (64.1)	111 (75.0)	221 (59.7)
- Female	186 (35.9)	37 (25.0)	149 (40.3)
- Total	518 (100)	148 (100)	370 (100)
Age at transplantation (years)			
- Mean ± SD	48.3 ± 13.9	53.1 ± 13.0	46.4 ± 13.8
- Median	49.6	54.9	46.7
- Range	7–78	17–76	7–78
Age at follow up			
- Mean ± SD	59.1 ± 12.8	66.2 ± 9.8	56.1 ± 12.7
- Median	60.4	67.5	57
- Range	24–85	36–85	24–84
Phototype			
- I, II	155 (29.9)	147 (99.3)	8 (2.2)
- III, IV	342 (66.0)	1 (0.7)	341 (92.2)
- V, VI	8 (1.5)	0	8 (2.2)
- NA	13 (2.5)	0	13 (3.5)
Chronic Actinic damage			
- Yes	137 (26.4)	82 (55.4)	55 (14.9)
- No	381 (73.6)	66 (44.6)	315 (85.1)
Solar lentigo			
- Yes	246 (47.5)	87 (58.8)	159 (43)
- No	249 (48.1)	51 (34.4)	198 (53.5)
- NA	23 (4.4)	10 (6.8)	13 (3.5)
Seborrheic keratosis			
- Yes	328 (63.3)	102 (68.9)	226 (61.1)
- No	174 (33.6)	39 (26.4)	135 (36.5)
- NA	16 (3.1)	7 (4.7)	9 (2.4)
Warts			
- Yes	78 (15.0)	31 (20.9)	47 (12.7)
- No	423 (81.7)	109 (73.6)	314 (84.9)
- NA	17 (3.3)	8 (5.4)	9 (2.4)
Cherry angioma			
- Yes	234 (45.2)	74 (50.0)	160 (43.2)
- No	275 (53.1)	71 (48.0)	204 (55.1)
- NA	9 (1.7)	3 (2.0)	6 (1.7)
Kidney disease			
- immune-mediated	86 (16.6)	17 (11.5)	69 (18.6)
- congenital	111 (21.4)	33 (22.3)	78 (21.1)
- acquired (not immune-mediated)	129 (24.9)	40 (27.0)	89 (24.1)
- NA	192 (37.1)	58 (39.2)	134 (36.2)

NA = not available.

**Table 2 medicina-55-00279-t002:** Life habits in our cohort of patients.

	N° Patients (%)	With Skin Cancer (%)	Without Skin Cancer (%)
Sun exposure			
- Never	32 (6.2)	7 (4.7)	25 (6.8)
- Occasional	376 (72.6)	104 (70.3)	272 (73.5)
- Frequent	98 (18.9)	30 (20.3)	68 (18.4)
- NA	12 (2.3)	7 (4.7)	5 (1.3)
UV-lamp exposure			
- Yes	39 (7.5)	8 (5.4)	31 (8.4)
- No	467 (90.2)	135 (91.2)	332 (89.7)
- NA	12 (2.3)	5 (3.4)	7 (1.9)
Previous sunburns			
- Yes	225 (43.4)	75 (50.7)	150 (40.5)
- No	277 (53.5)	66 (44.6)	211 (57)
- NA	16 (3.1)	7 (4.7)	9 (2.5)
Use of sunscreen			
- Yes	270 (52.1)	77 (52)	193 (52.2)
- No	189 (36.5)	54 (36.5)	135 (36.5)
- NA	59 (11.4)	17 (11.5)	42 (11.3)
Outdoor job			
- Yes	138 (26.6)	44 (29.7)	94 (25.4)
- No	360 (69.5)	100 (67.6)	260 (70.3)
- NA	20 (3.9)	4 (2.7)	16 (4.3)

NA = not available.

**Table 3 medicina-55-00279-t003:** Cox regression analysis.

	B	SE	*p*-Value	Exp (B)	95% CI for Exp (B)
Gender (male vs. female)	0.78	0.19	**<0.0001**	2.18	1.49–3.20
Age at transplantation (years)	0.08	0.008	**<0.0001**	1.08	1.06–1.10
Age at follow up (years)	0.05	0.008	**<0.0001**	1.05	1.04–1.07
Phototype (I, II vs. IIII, IV, V, VI)	5.72	1.00	**<0.0001**	306.40	42.84–2191.32
Chronic Actinic Damage (yes vs. no)	0.78	0.17	**<0.0001**	2.18	1.57–3.03
Solar lentigo (yes vs. no)	0.38	0.18	**0.03**	1.46	1.03–2.08
Seborrheic keratosis (yes vs. no)	0.14	0.19	0.46	0.87	0.60–1.26
Warts (yes vs. no)	0.11	0.21	0.60	1.11	0.74–1.68
Cherry angioma (yes vs. no)	0.15	0.17	0.37	1.16	0.84–1.61
Kidney disease					
(congenital vs. immune mediated)	0.53	0.31	0.09	1.70	0.93–3.12
(acquired, not immune-mediated vs. immune mediated)	0.55	0.30	0.06	1.74	0.97–3.12
Sun exposure (occasional vs. never) (frequent vs. never)	0.33	0.40	0.40	1.39	0.64–3.03
UV-lamp exposure (yes vs. no)	0.29	0.37	0.46	1.33	0.64–2.77
Previous sunburns (yes vs. no)	0.45	0.17	**0.01**	1.57	1.13–2.20
Use of sunscreen (yes vs. no)	0.18	0.18	0.30	1.20	0.84–1.71
Outdoor job (yes vs. no)	0.17	0.18	0.35	1.19	0.83–1.70

B, regression coefficients; SE, standard error; Exp, exponential function.

**Table 4 medicina-55-00279-t004:** Schematic representation of the number of skin cancers developed by our cohort of patients during follow-up.

	Patients		Total
N. of Tumors	M (%)	F (%)	N (%)
0	221 (66.6)	149 (80.1)	370 (71.4)
1	53 (16.0)	24 (12.9)	77 (14.9)
2	23 (6.9)	8 (4.3)	31 (6.0)
3	19 (5.7)	2 (1.1)	21 (4.0)
4	16 (4.8)	3 (1.6)	19 (3.7)
Total	332 (100)	186 (100)	518 (100)

M = male; F = female.

**Table 5 medicina-55-00279-t005:** Logistic regression analysis. Odds ratio and 95% CI.

	Single Tumor (Patients = 77)	Multiple Cancers (Patients = 71)	OR (95% CI)
Gender			
Male	53 (69%)	58 (82%)	2.0 (0.9–4.4)
Female	24 (31%)	13 (18%)	1 (ref)
Actinic Damage			
Yes	29 (38%)	52 (73%)	4.8 (2.3–9.8)
No	48 (62%)	18 (25%)	1 (ref)
NA	-	1 (1%)	
Solar lentigo			
Yes	36 (47%)	51 (72%)	3.7 (1.7–7.8)
No	37 (48%)	14 (20%)	1 (ref)
NA	4 (5%)	6 (8%)	
Seborrheic keratosis			
Yes	51 (66%)	51 (72%)	1.3 (0.6–2.8)
No	22 (29%)	17 (24%)	1 (ref)
NA	4 (5%)	3 (4%)	
Warts			
Yes	13 (17%)	18 (25%)	1.9 (0.8–4.3)
No	61 (79%)	48 (68%)	1 (ref)
NA	3 (4%)	5 (7%)	
Cherry angioma			
Yes	37 (48%)	37 (52%)	1.2 (0.6–2.4)
No	39 (51%)	32 (45%)	1 (ref)
NA	1 (1%)	2 (3%)	

NA = not available.

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
