# Peer review of "Identification of Risk Factors for Multiple Non-Melanoma Skin Cancers in Italian Kidney Transplant Recipients"

_medicina, 2019, doi:10.3390/medicina55060279_

Reviewer 1 Report

Review of:

Manuscript ID: medicina-440068

Title: Identification of risk factors for multiple Non-Melanoma Skin Cancers in Italian kidney transplant recipients

Dear Editor,

Thank you for sending me this article to review. This manuscript reports on a study of non-melanoma skin cancer in a cohort of kidney transplant recipients. While this study does address an interesting and important question, there are a number of improvements that should be made before this paper would be ready for publication. This is not an extensive list and the authors should be prepared to thoroughly review their paper.

1.      The paper would be better presented if the methods section is moved to between the introduction and the results.

2.      The authors need to review table layout guidelines, I am not aware of table guidelines that encourage so many horizontal rules.

3.      My biggest concern with the paper is the description and reporting of the statistical analyses. The statistical methods section needs to be expanded to include all statistical analyses undertaken. I found that the statistical methods and the results section did not flow and it was hard to follow what was being presented. Under the heading “Risk factors for keratinocyte neoplasms” there is a sentence “results of clinical data are reported in Table 1 and2” which is after Table 3 has been introduced. I have not been detailed with the improvements needed – it would be too time consuming and I feel that the whole of both sections needs completely rewritten with a more logical approach.

4.      In the discussion, what do the authors mean by “ in our experience”. Is this referring to the results of the study or just their anecdotal evidence? The limitations section needs to be expanded – I could not really find one at all – to include the potential for recall bias, misclassification bias to impact on the findings. Also, the fact that some patients with incomplete visit profiles were excluded from the analysis needs to be addressed – did they differ from those with complete profiles?

5.      Finally the paper needs a thorough proof-read. There are a number of instances in which the writing could be improved or corrected. Specifically, (for example) could the authors in the Abstract address the sentence “two skin cancers in 31 patients, 3 in 21, whereas…” 3 in 21 needs that word patients after it. In the Tables and the manuscript, there are instances of IC95%. Should this be CI95%? I would prefer all instances to be changed to 95% CI. There are also a number of instances of inadequate reference reporting in the paper – eg Terrsari and coll. I have never seen a reference reported this way,

Author Response

Medicina

Manuscript ID 440068

Title: Identification of risk factors for multiple Non-Melanoma Skin Cancers in Italian kidney transplant recipients

Dear Editor,

We have read with interest the Reviewers’comments and we have prepared the present letter with a point-by-point response to the questions.

Furthermore, all the changes and revisions are reported in the revised version of the manuscript with the “track changes” system.

We would like to thank the referres for the constructive criticism. Please find enclosed  answers to their questions. We do hope that now the manuscript could be accepted for publication in Medicina.

Reviewer 1

Dear Editor,

Thank you for sending me this article to review. This manuscript reports on a study of non-melanoma skin cancer in a cohort of kidney transplant recipients. While this study does address an interesting and important question, there are a number of improvements that should be made before this paper would be ready for publication. This is not an extensive list and the authors should be prepared to thoroughly review their paper.

1.      The paper would be better presented if the methods section is moved to between the introduction and the results.

Re. We agree with your comment, although the structure of the submitted paper was in accordance with the Author Guidelines of Medicina jounal. In its revised form, the Methods section is now located between the Introduction and the Results.

2.      The authors need to review table layout guidelines, I am not aware of table guidelines that encourage so many horizontal rules.

Re. We have checked the Author Guidelines, but no recommendation regarding table layout are present, and, in our study, we have examined so many risk factors that a plenty of data deserve to be reported in the Tables. Notwithstanding we agree with the comment, mainly for the Table 1. In conclusion, we believe that, if the Reviewer would consider to be strictly necessary, we coud delete the risk factors that did not result significant (i.e. seborrheic keratosis, kind of kidney disease..)

3.      My biggest concern with the paper is the description and reporting of the statistical analyses. The statistical methods section needs to be expanded to include all statistical analyses undertaken. I found that the statistical methods and the results section did not flow and it was hard to follow what was being presented. Under the heading “Risk factors for keratinocyte neoplasms” there is a sentence “results of clinical data are reported in Table 1 and2” which is after Table 3 has been introduced. I have not been detailed with the improvements needed – it would be too time consuming and I feel that the whole of both sections needs completely rewritten with a more logical approach.

Re.Thank you for your comment. We have carefully checked the Methods (Statistical analysis) and Results sections and made many changes, following a more logical approach. We believe that now those sections are flowing and easy to follow. Concerning the mentioned sentence dealing with the data reported in Table 1 and 2, on the basis of the rearrangements of the results, we decided to delete it.

4.      In the discussion, what do the authors mean by “ in our experience”. Is this referring to the results of the study or just their anecdotal evidence?

The limitations section needs to be expanded – I could not really find one at all – to include the potential for recall bias, misclassification bias to impact on the findings. Also, the fact that some patients with incomplete visit profiles were excluded from the analysis needs to be addressed – did they differ from those with complete profiles?

Re. The words “in our experience” might be changed into “in clinical practice”. Indeed, we aimed to highlight not only our personal experience in taking care of OTRs referring to the Dermatology Outpatients Clinic, but also the fact that, to date, risk factors for the development of multiple NMSCs have not yet been fully reported. In fact, NMSCs are extremely frequent in OTRs, but also in general population. It seems to be of great important to recognize which risk factors, if any, could determine the development of numerous skin cancers in this category of subjects.

The limitation section is located at the end of the Discussion (last two paragraphs), and we have added some sentences regarding bias, and to properly explain the reason why patients with incomplete profiles were excluded.

5.      Finally the paper needs a thorough proof-read. There are a number of instances in which the writing could be improved or corrected. Specifically, (for example) could the authors in the Abstract address the sentence “two skin cancers in 31 patients, 3 in 21, whereas…” 3 in 21 needs that word patients after it. In the Tables and the manuscript, there are instances of IC95%. Should this be CI95%? I would prefer all instances to be changed to 95% CI. There are also a number of instances of inadequate reference reporting in the paper – eg Terrsari and coll. I have never seen a reference reported this way,

Re. The required change in the Abstract has been edited. Furthermore we found some mistakes in the text, that were correted.

IC95% and CI95% stands for Confidence Interval 95% and has been changed, as required (95% CI).

We have carefully checked the references and their report in the paper: “Tessari and coll.” and “Gogia and coll.” are now changed in “Tessari et al.” and “Gogia et al”.

We have also added three new articles (number 5, 42 and 43).

Regarding the ref Tessari et al. (now number 36), the Authors reported “a increased long-term probability of developing a subsequent NMSC compared with patients with no previous NMSC “ and also that “none of the clinical risk factors for the first NMSC could predict the onset of the second NMSC...” They also reported  high presence of AKs in their cohort of patients developing 1 BCC/SCC. Since we believe that this sentences could support our Discussion, and in order to better explain such concept, we edited this paragraph, as reported in the revised manuscript.

Reviewer 2 Report

In this study the authors conducted an analysis on kidney transplant recipients to determine the rate of non-melanoma skin cancer (NMSC) in these patients and determine the risk factors which played a role in NMSC incidence in this cohort.

Major Points:

In the results section "skin phototype groups" were discussed.  Can you please define either in the methods section or results what this means.

Table titles all need to be made clearer.  They were very vague and hard to understand.

I would personally like to see the influence of gender on risk analyzed as well.  For example do females have more or less warts or cherry angiomas.  This might lead to a more in depth analysis of the data.

The prose section of the results is lacking.  There are many large tables but no prose to help the reader understand the highlights the authors wish to show.  Please expand on the results and describe the main points each table is showing for the reader.

For Table 3, what is the time frame of incidence?  Please make that more clear.

In the Discussion, what about the rapalog data?  I have seen that the treatment of transplant recipients with rapalogs actually decreases NMSC incidence.  I would appreciate a bit of discussion of these and how they fit into your model.

Minor Points:

Please revise the paper for grammar.  There are many writing errors. 

Please fix and coll. to et. al in the Discussion.

Author Response

Medicina

Manuscript ID 440068

Title: Identification of risk factors for multiple Non-Melanoma Skin Cancers in Italian kidney transplant recipients

Dear Editor,

We have read with interest the Reviewers’comments and we have prepared the present letter with a point-by-point response to the questions.

Furthermore, all the changes and revisions are reported in the revised version of the manuscript with the “track changes” system.

We would like to thank the referres for the constructive criticism. Please find enclosed  answers to their questions. We do hope that now the manuscript could be accepted for publication in Medicina.

Reviewer 2

In this study the authors conducted an analysis on kidney transplant recipients to determine the rate of non-melanoma skin cancer (NMSC) in these patients and determine the risk factors which played a role in NMSC incidence in this cohort.

Re. Thank you for your comment. First of all, we want to highlight that the aim of our work was to determine the risk factors for the development of multiple NMSCs in our cohort.  We will review the paper according to your comments (see above).

Major Points:

In the results section "skin phototype groups" were discussed. Can you please define either in the methods section or results what this means.

Re. The “skin phototype groups” were assessed according to the Fitzpatrick classification. The meaning has been added either in the Material section or in the Results section. Accordingly, the corresponding reference has been added (number 5). The reference numbering has been changed accordingly.

Table titles all need to be made clearer.  They were very vague and hard to understand.

Re. The table titles have been edited according to the requirement.

I would personally like to see the influence of gender on risk analyzed as well.  For example do females have more or less warts or cherry angiomas.  This might lead to a more in depth analysis of the data.

Re. In our study cohort we have not find a statistically significant association between the presence of warts and/or cherry angioma and gender, but only an association at the limit of sugnificance between gender and actinic damage. All models were adjusted by age at transplantation and gender as potential confounding variables in all the analysis.

The prose section of the results is lacking.  There are many large tables but no prose to help the reader understand the highlights the authors wish to show.  Please expand on the results and describe the main points each table is showing for the reader.

Re. As stated above (see answers to the Reviewr 1), the Results section has been largely rearranged following a more logical approach. In our opinion, now the manuscript is more fluid in reading and our results are highlighted, even the tables show the more important data to the reader.

Regarding the Tables size, we believe that, if the Reviewer would consider to be strictly necessary, we coud delete the risk factors that did not result significant (i.e. seborrheic keratosis, kind of kidney disease..)

For Table 3, what is the time frame of incidence?  Please make that more clear.

Re. As stated in the Results, Table 3 shows the number of NMSCs developed in our cohort during the follow-up. In order to stress the time frame, we added the words “during follow up” in the Table title.

In the Discussion, what about the rapalog data?  I have seen that the treatment of transplant recipients with rapalogs actually decreases NMSC incidence.  I would appreciate a bit of discussion of these and how they fit into your model.

Re. We added a short discussion about Rapalogs in the Discussion, and further references (number 42-43).

Minor Points:

Please revise the paper for grammar.  There are many writing errors. 

Re. We apologize for the writing errors, now we have carefully revised our paper for grammar. Corrections are reported in the revised manuscript.

Please fix and coll. to et. al in the Discussion.

Re. The changes in the Discussion have beed edited accordingly

Round  2

Reviewer 1 Report

Dear Editor and Authors, 

My concerns have been addressed by the authors.

Author Response

Thank you for your comments

Reviewer 2 Report

Thank you for addressing the issues described in my previous comments.

Author Response

Thank you for your comments